# Advancements in Materials for 3D-Printed Microneedle Arrays: Enhancing Performance and Biocompatibility

**DOI:** 10.3390/mi15121433

**Published:** 2024-11-28

**Authors:** Mahmood Razzaghi, Joel Alexander Ninan, Mohsen Akbari

**Affiliations:** 1Laboratory for Innovations in Microengineering (LiME), Department of Mechanical Engineering, University of Victoria, Victoria, BC V8P 5C2, Canada; joelninan@uvic.ca; 2Terasaki Institute for Biomedical Innovations, Los Angeles, CA 90050, USA

**Keywords:** 3D-printed microneedle arrays, biocompatible materials, photopolymer resins, drug delivery, biomedical applications

## Abstract

The rapid advancement of 3D printing technology has revolutionized the fabrication of microneedle arrays (MNAs), which hold great promise in biomedical applications such as drug delivery, diagnostics, and therapeutic interventions. This review uniquely explores advanced materials used in the production of 3D-printed MNAs, including photopolymer resins, biocompatible materials, and composite resins, designed to improve mechanical properties, biocompatibility, and functional performance. Additionally, it introduces emerging trends such as 4D printing for programmable MNAs. By analyzing recent innovations, this review identifies critical challenges and proposes future directions to advance the field of 3D-printed MNAs. Unlike previous reviews, this paper emphasizes the integration of innovative materials with advanced 3D printing techniques to enhance both the performance and sustainability of MNAs.

## 1. Introduction

Microneedle arrays (MNAs) are medical devices with micron-scaled needles to administer vaccines, drugs, and other therapeutic agents [1]. Additionally, MNAs have shown significant potential and successfully collected skin interstitial fluid (ISF) for detection purposes [2]. MNAs are designed to penetrate the outermost layer of the skin, the stratum corneum, without reaching the pain receptors located deeper within the dermis. This unique capability allows microneedles (MNs) to deliver therapeutic agents directly into the epidermis and dermis, facilitating efficient and pain-free administration of medications, vaccines, and other bioactive substances [3].

The significance of MNAs is underscored by their potential to improve patient compliance, reduce the risk of infection associated with traditional hypodermic needles, and provide a platform for self-administration of treatments [4]. For instance, dissolving MNAs have been demonstrated to deliver monoclonal antibodies more effectively than subcutaneous injections, showing enhanced therapeutic effects in osteoporosis models [5]. Additionally, MNAs have applications in diagnostic monitoring and sampling of interstitial fluid, making them a versatile tool in modern healthcare [6,7]. Numerous studies have investigated the application of 3D-printed MNAs in both drug delivery [8,9,10,11] and diagnostic [12,13,14] fields. Notably, customizable photothermal MNAs produced using low-cost 3D printers have been shown to effectively treat bacterial infections via intradermal photothermal therapy [15].

Traditional manufacturing methods for fabricating MNAs include techniques such as micro-milling (using either wet or dry cutting) [16], etching [17,18], photolithography (involving thermal or photo-polymerization) [19,20], molding approaches [21,22], cleanroom-free molding [23], injection molding [24], laser patterning [25], lithography [26,27], and photolithography with an elastic-capillarity-driven self-assembly mechanism [28]. Many of these conventional techniques are time-consuming and require multiple stages, limiting their cost-efficiency. As a result, more affordable and accessible manufacturing technologies are needed for MNA production, and 3D printing has emerged as a modern alternative with several advantages over traditional methods. For example, high-resolution 3D printing has enabled the production of latticed MNAs with enhanced cargo loading and tunable release kinetics for therapeutic applications [29]. Improvements in 3D printing resolution, feature precision, and the availability of cost-effective materials have made it possible to create various types of MNAs [16,30,31,32]. This technology allows for the fabrication of complex structures with precise control over geometry, shape, size, and mechanical and biological properties [33,34,35,36]. Several 3D printing techniques, including stereolithography (SLA) [30,33,37,38,39,40,41,42,43,44,45,46,47], digital light processing (DLP) [44,48,49,50,51,52], fused deposition modeling (FDM) [53,54,55,56], and two-photon polymerization (2PP) [57,58,59,60,61,62] have been utilized to manufacture MNAs.

The choice of material is a critical factor in the design and performance of 3D-printed MNAs. The material must exhibit two key properties: biocompatibility and mechanical strength [63]. Recent work highlights that the use of biocompatible photocurable resins in MNAs not only provides mechanical durability but also reduces environmental impact, aligning with sustainability goals [5]. It must have also an appropriate degradation profile if biodegradable MNAs are desired. Biocompatibility ensures that the MNAs do not provoke an adverse immune response when inserted into the skin [64]. Mechanical strength is essential to ensure that the MNAs can penetrate the skin without bending or breaking [65]. Additionally, for applications involving drug delivery, the materials must be able to encapsulate and release therapeutic agents in a controlled manner [66].

Recent advancements in material science have led to the development of novel polymers, hydrogels, and composites specifically tailored for the 3D printing of MNAs. For instance, gelatin methacryloyl (GelMA)-based MNAs produced using vat photopolymerization demonstrated high-resolution geometries and effective transdermal drug delivery with minimal mechanical failure [67]. Materials such as polylactic acid (PLA) [56,68] and polyglycolic acid (PGA) [69,70] have been frequently used due to their favorable properties. Innovations in material formulations have also enabled the creation of MNAs that can dissolve or degrade after insertion, offering a safe and efficient method for drug delivery. These advancements in materials are crucial for enhancing the performance and biocompatibility of MNAs, thereby broadening their potential applications in clinical settings [71].

This review aims to provide a comprehensive overview of the recent advancements in materials used for 3D-printed MNAs, with a focus on enhancing their performance and biocompatibility. While earlier reviews have explored the general advancements in 3D-printed MNAs, this review uniquely focuses on the intersection of material science and emerging trends like personalized medicine and stimuli-responsive materials. The objectives are to: (a) summarize the current state of MNA technology and the role of 3D printing in its development; (b) evaluate the various materials that have been utilized in the 3D printing of MNAs, highlighting their properties, advantages, and limitations; (c) discuss the impact of material choice on the performance, biocompatibility, and application potential of MNAs; and (d) identify the key challenges and future directions in the field, particularly in relation to material innovation and 3D printing techniques.

The scope of this review encompasses an analysis of recent studies that illustrate the practical applications of advanced materials in 3D-printed MNAs. By synthesizing this information, the review aims to provide insights that can guide future research and development efforts, ultimately contributing to the advancement of MNA technology and its integration into mainstream medical practice.

In conclusion, the intersection of 3D printing technology and material science holds significant promise for the future of MNAs. Through ongoing research and innovation, it is possible to develop MNA systems that are not only more effective and reliable but also more accessible and patient-friendly. This review seeks to illuminate the path forward, highlighting the critical role of advanced materials in realizing the full potential of 3D-printed MNAs. Figure 1 shows a schematic illustrating the 3D printing process for fabricating MNAs, including the preparation of the 3D printing material, one of the main steps of the MNA 3D printing process.

## 2. Types of Materials Used in 3D-Printed Microneedle Arrays

As stated, the materials used in 3D-printed MNAs play a crucial role in determining their mechanical strength, biocompatibility, and overall performance. This section discusses the various types of materials utilized, including photopolymer resins, biodegradable polymers, hydrogels, composite resins, and innovative materials. This review examines a variety of materials, including photopolymer resins, biodegradable polymers, hydrogels, and emerging materials, used in the fabrication of 3D-printed MNAs, with a focus on addressing underexplored aspects highlighted in previous reviews.

### 2.1. Photopolymer Resins

Photopolymer resins are widely used in 3D printing due to their ability to rapidly solidify upon exposure to specific wavelengths of light. These materials offer high resolution and the ability to create intricate structures, making them suitable for MNA fabrication [72]. Photopolymer resins can be broadly categorized into ultraviolet (UV)-curable resins and visible light-curable resins.

UV-curable resins have been used for 3D printing of MNAs [73,74]. They solidify when exposed to UV light, a process that involves photoinitiators that start a polymerization reaction. These resins are popular in SLA and DLP 3D printing technologies. Figure 2 shows a schematic representation of the fabrication of GelMA MNs using a UV-curable resin and specifically what happens to the resin in the 3D printing process of photocrosslinking of GelMA hydrogel during 3D printing [67]. UV-curable resins offer several advantages, including their high resolution, capability of producing fine details, which is essential for MNAs, and their good mechanical properties. They can be engineered to provide the necessary mechanical strength for skin penetration without breaking and are versatile. UV-curable resins can be modified with various additives to enhance their properties [16,75].

However, one challenge is ensuring the biocompatibility of UV-curable resins. Residual monomers and photoinitiators can cause cytotoxicity, which necessitates thorough post-processing to remove any unreacted components [76].

Visible light-curable resins use light in the visible spectrum to initiate polymerization. These resins are particularly advantageous for applications requiring less intensive curing conditions compared to UV light. Visible light-curable resins are safer and more user-friendly because of the reduced risk of exposure to harmful UV radiation [77]. They are also effective for certain polymers and enable the use of materials that might degrade under UV light. Despite these benefits, visible light-curable resins may have limitations in terms of depth of cure and overall mechanical properties compared to their UV counterparts [77].

### 2.2. Biodegradable Polymers

Biodegradable polymers are designed to break down within the body over time, which is beneficial for drug delivery applications where the MNAs dissolve after releasing their payload. Common biodegradable polymers used in 3D printing MNAs include: PLA [56,68], which is known for its biocompatibility and biodegradability and is a popular choice that can be processed using various 3D printing techniques, providing adequate mechanical strength for MNAs; polyglycolic ccid (PGA) [69,70], which degrades faster than PLA and is often used in conjunction with other polymers to tailor degradation rates; and poly(lactic-co-glycolic acid) (PLGA) [78,79,80], which is a combination of PLA and PGA that offers customizable degradation rates and mechanical properties and is widely used in drug delivery systems.

Biodegradable polymers are advantageous as they eliminate the need for removal after administration. However, controlling the degradation rate to match the desired therapeutic profile remains a challenge [81]. The use of biodegradable polymeric materials further enables drug encapsulation within the needle matrix and release into the skin upon degradation or diffusion. Figure 3 shows a schematic workflow for the fabrication of poly(propylene fumarate-co-propylene succinate) (PPFPS) (a biodegradable polymer) MNAs and their in vivo drug delivery evaluation [82].

### 2.3. Hydrogels

Hydrogels are hydrophilic polymer networks capable of absorbing and retaining significant amounts of water, making them particularly effective for drug delivery due to their ability to encapsulate and release drugs in a controlled manner [83]. In 3D-printed MNAs, hydrogels can be categorized into natural and synthetic types. Natural hydrogels, such as alginate [84,85] and gelatin [86,87], are inherently biocompatible and can be chemically modified to enhance their properties. On the other hand, synthetic hydrogels like polyethylene glycol diacrylate (PEGDA) [12,88] offer precise control over mechanical properties. The softness and flexibility of hydrogels make them ideal for creating MNAs that minimize discomfort during insertion. However, their mechanical strength is typically lower than that of solid polymers, which may limit their effectiveness in applications requiring deeper skin penetration [89].

As mentioned, the structure of hydrogels can absorb and retain water, enabling efficient encapsulation and controlled release of drugs. This makes them ideal for transdermal drug delivery, ensuring minimal discomfort and sustained therapeutic effects. Furthermore, their biocompatibility reduces adverse reactions, enhancing patient safety [83]. In biosensing, hydrogels’ ability to integrate with biomolecules and respond to environmental stimuli, such as pH or glucose levels, allows precise, real-time monitoring, making them indispensable in advanced biomedical applications [12].

Research has demonstrated that 3D-printed gelatin GelMA hydrogel MNAs possess customizable heights and sharpness, achieving exceptional mechanical performance without breakage under a displacement of up to 0.3 mm. These findings highlight GelMA MNAs as mechanically robust and highly suitable for transdermal drug delivery, providing precise control over geometry and mechanical properties with high-resolution fabrication [67].

### 2.4. Composite Resins and Materials

Composite materials combine two or more distinct materials to harness their individual strengths while mitigating their weaknesses. In the context of 3D-printed MNAs, composites often involve a mixture of polymers with inorganic materials or other polymers [90,91]. Composite materials allow for the customization of MNAs’ mechanical, chemical, and biological properties, making them highly versatile for various applications. However, achieving uniform dispersion of components and preventing phase separation are common challenges in fabricating composite materials.

A recent study [92] presented an innovative 3D-printed MNA utilizing a composite hydrogel ink made from four monomers: 2-(dimethylamino)ethyl methacrylate, N-isopropylacrylamide, acrylic acid, and acrylamide. These monomers were crosslinked using aluminum hydroxide nanoparticles, yielding a material with outstanding mechanical strength suitable for precise 3D printing. The printed MNAs showcased customizable geometries and triple responsiveness to pH, temperature, and glucose. Moreover, their effectiveness for cytotoxicity-free transdermal drug delivery was confirmed using bovine serum albumin as a model drug. This innovative approach highlights the potential of advanced composite formulations for developing multifunctional MNAs in biomedical applications.

### 2.5. Innovative Materials

Ongoing research is continually uncovering new materials with potential applications in 3D-printed MNAs. Innovative materials aim to push the boundaries of performance, biocompatibility, and functionality. Some of these materials are smart polymers, also known as stimuli-responsive polymers, that change their properties in response to environmental stimuli such as temperature, pH, or light [93]. Conductive polymers [94] can be used to fabricate MNAs that not only deliver drugs but also have sensing capabilities in applications such as biosensing. Nanocomposites, which incorporate nanoparticles into polymer matrices, can significantly enhance the properties of MNAs like silver nanoparticles by providing antibacterial properties, which are beneficial for preventing infections [95]. Additionally, 3D-Printable Bioceramics like hydroxyapatite and bioactive glass are emerging as potential materials for MNAs, particularly in bone-related applications. These materials offer biocompatibility and osteoconductivity, promoting bone growth and integration [63]. Innovative materials hold the promise of expanding the capabilities of MNA technology, enabling new applications and improving existing ones. However, challenges such as ensuring consistent material properties, scalability of production, and cost-effectiveness need to be addressed.

The emerging trend of 4D printing in the fabrication of MNAs is made possible using specific materials. It refers to an advanced form of 3D printing where the printed object can change its shape, properties, or behavior over time in response to external stimuli, such as temperature, moisture, or light. This transformation occurs due to the materials used, which are designed to adapt and reshape themselves in a programmable manner after the initial printing process. In 4D printing, the fourth dimension is time, as the printed structure undergoes a change post-fabrication, allowing for dynamic, responsive applications. Han et al. [88] applied 4D printing to fabricate a bioinspired MNA with backward-facing barbs for enhanced tissue adhesion. This technology enabled the precise creation of MNAs with complex geometries, including curved barbs that are challenging to produce using traditional methods. Through the use of projection microstereolithography (PµSL), a digital light processing technique, these MNAs were formed with programmable shape transformations that allow the barbs to curve backward upon desolvation, significantly improving tissue adhesion. The material used in this process, specifically the photocurable resin (PEGDA 250), played a key role in the MNAs’ functionality. By adjusting the crosslinking density during the printing process, the barbs were able to bend and create the desired shape. This controlled material composition allowed for the fabrication of a highly effective MNA, demonstrating 18 times stronger tissue adhesion compared to conventional MNAs. Furthermore, drug release tests showed sustained delivery, highlighting the potential for long-term applications in drug delivery and biosensing. Figure 4 shows their schematic illustration of the 4D printing approach to program deformation of horizontally printed barbs into a backward-facing shape. C–E) SEM images of 4D-printed MNA with backward-facing barbs [88].

The choice of material for 3D-printed MNAs is crucial for their performance, safety, and applicability. Photopolymer resins, biocompatible materials, composite materials, and innovative materials each offer unique benefits and challenges. Understanding the properties and potential of these materials allows for the design of MNAs tailored to specific medical applications. As material science and 3D printing technologies continue to evolve, the development of advanced MNAs will undoubtedly enhance their clinical utility and broaden their impact on healthcare.

## 3. Material Properties and Their Impact on the Performance of Microneedle Arrays

As discussed earlier, the performance and efficacy of MNAs are highly dependent on the materials used in their fabrication. Critical material properties that influence their performance include mechanical properties, biocompatibility, degradation and stability, and drug loading and release characteristics. Understanding these properties is essential for optimizing MNA design and ensuring safe and effective drug delivery. Here, the material properties and their effect on the performance of MNA are discussed.

### 3.1. Mechanical Properties

The mechanical properties of MNA materials are crucial for ensuring successful skin penetration and structural integrity during application. The primary mechanical properties of interest are strength, hardness, elasticity, and fracture toughness. These properties determine the ability of MNAs to withstand the forces exerted during insertion into the skin without bending, breaking, or causing discomfort to the patient. MNAs must possess sufficient strength and hardness to penetrate the stratum corneum, the tough outermost layer of the skin. Some materials may have sufficient strength, but they may lack the necessary elasticity and fracture toughness, leading to potential brittleness [65].

The mechanical properties of MNAs can be enhanced through various strategies, such as material type, adjusting the crosslinking density, crystallinity, and molecular weight of individual polymers, incorporating reinforcing agents to form polymer complexes, adjusting the 3D printing process, and optimizing the geometric parameters of the MNAs [65]. Various studies have aimed to develop MNAs with sufficient mechanical properties and have explored the mechanical characteristics of 3D-printed MNAs. Baykara et al. [67] investigated the effect of UV exposure times of the 3D printing process on the mechanical properties of GelMA MNAs. Their results showed that increasing the exposure time improved the mechanical properties of MNAs. Rajesh et al. [29] evaluated the mechanical properties of latticed MNAs with different shapes using finite element analysis (FEA). Their results showed that square-based 2-tier and 3-tier designs demonstrated higher mechanical integrity scores compared to other shapes. These designs withstood the necessary insertion forces for skin penetration without fracturing, ensuring mechanical robustness [29]. In another research, the effect of needle cross-section shape on penetration force was studied. In this research, PEGDA-based MNAs with round, hexagonal, square, and triangular cross-section shapes were 3D printed. Based on the results, MNAs with a hexagonal cross-section shape had a lower penetration force as an advantage [12].

The parameters of the 3D printing process can also influence the mechanical properties of 3D-printed MNAs. It was shown that the 3D printing tilt angle can affect the force for penetration into the skin. The results of this study showed that the minimum puncture force was achieved using a 45° printing tilt angle. Using this angle, the puncture force was reduced by 38% compared to MNAs printed with a tilting angle of 0° [96]. Figure 5A shows 3D-printed PEGDA-based MNAs with varying needle cross-sectional shapes, designed to evaluate the impact of needle shape on mechanical properties [12] and Figure 5B shows two different designs for 3D-printed MNAs with different tilt angles (0° and 45° in two directions) to investigate the effect of printing tilt angle on the mechanical properties [96].

### 3.2. Biocompatibility

Biocompatibility is a critical requirement for materials used in MNAs to ensure that they do not provoke an adverse biological response when inserted into the skin. Biocompatibility encompasses several factors, including cytotoxicity, immunogenicity, and potential for causing allergic reactions or inflammation. For cytotoxicity, materials used in MNAs should be non-toxic to human cells. Metals like stainless steel and titanium are known for their excellent biocompatibility and are widely used in medical devices [97]. Polymers such as PLA, PGA, and polycaprolactone (PCL) are biodegradable and biocompatible, making them suitable for MNAs that degrade in the body over time [98]. For immunogenicity, the material should not elicit an immune response that could lead to inflammation or rejection [99]. Hydrogels, such as those made from hyaluronic acid (HA) are highly biocompatible and have been used successfully in dissolvable MNA systems [100]. Furthermore, to avoid allergic reactions, materials must be free from allergens and other harmful substances. Polymers and hydrogels derived from natural sources or synthesized to mimic natural substances are less likely to cause allergic responses compared to synthetic materials [101].

One of the significant challenges in achieving biocompatibility of 3D-printed MNAs lies in the presence of residual monomers and photoinitiators after the 3D printing process. These unreacted components can potentially cause cytotoxic effects, posing risks to the safety and efficacy of the MNAs in biomedical applications [76]. To overcome this issue, thorough post-processing techniques are essential. These methods aim to remove any residual monomers and photoinitiators, thereby minimizing cytotoxicity and enhancing the biocompatibility of the final product. Strategies such as solvent washing, UV curing, and thermal treatments have been explored to effectively mitigate this challenge.

The biocompatibility and toxicity of various 3D-printed MNAs have been explored in numerous studies. Zhou et al. [92] investigated the biocompatibility and cytotoxicity of hydrogel-based MNAs, which were fabricated using a DLP 3D printer. Their findings revealed that these hydrogel MNAs did not exhibit any cytotoxic effects. Similarly, Monou et al. [102] assessed the biocompatibility of drug-coated MNAs, also 3D printed using a DLP printer. In this study, the MNAs were created with a commercial biocompatible resin known as Dental SG, produced by Formlabs. The MNAs were further coated with a drug using an extrusion 3D printer. The researchers evaluated the toxicity of these MNAs by performing cell studies, as well as histological and immunohistochemistry tests on human skin samples. The results confirmed that the MNAs were safe for transdermal use and did not induce any cytotoxic effects.

### 3.3. Degradation and Stability

The degradation and stability of MNA materials are essential for ensuring that they perform their intended function over the desired period. This property is particularly important for biodegradable MNAs designed to dissolve or degrade after delivering their payload. The rate at which an MNA degrades can be tailored depending on the application. Stability under physiological conditions is crucial for maintaining the structural integrity and functionality of MNAs until they have fulfilled their purpose. The stability of MNA materials can be influenced by environmental factors such as temperature, humidity, and pH. Careful consideration of these factors during material selection and design is necessary to ensure that the MNAs perform reliably under varying conditions [103].

Tailored degradation rates are essential for MNA materials to align with application needs. Rapid degradation is ideal for short-term drug release, ensuring timely therapeutic effects, while slow, controlled degradation supports sustained delivery over extended periods. Customizing degradation profiles enhances efficacy, minimizes side effects, and optimizes treatment outcomes for diverse medical applications. The use of hydrogels and biodegradable polymers allows precise control over degradation, aligning material properties with specific medical needs for diverse treatment scenarios [83].

### 3.4. Drug Loading and Release Characteristics

The drug loading and release characteristics of MNA materials are pivotal for achieving the desired therapeutic outcomes. These characteristics depend on the material’s ability to incorporate, protect, and release the drug at a controlled rate. The capacity of a material to load a sufficient amount of drug is critical for ensuring therapeutic efficacy. Hydrogels and certain polymers, such as chitosan and gelatin, have high drug-loading capacities due to their porous structures and ability to swell in aqueous environments [104]. Controlled and sustained release of drugs can be achieved by strategies like selecting appropriate materials and designing the MNA architecture [66]. The stability of the drug within the MNA material is crucial for maintaining its efficacy. Encapsulation within biocompatible and protective materials such as PLGA and polyvinylpyrrolidone (PVP) can shield the drug from degradation due to environmental factors such as light, moisture, and enzymatic activity [105]. Various mechanisms can be employed to trigger drug release from MNAs, including diffusion, degradation, and swelling-induced release [106].

## 4. Fabrication Techniques for 3D-Printed Microneedle Arrays

The utilization of 3D printing represents a relatively recent approach to creating MNAs, as it offers various advantages compared to traditional methods. Advances in printing resolution, precision of features, and the accessibility of cost-effective printing materials has enabled the production of diverse types of MNAs using 3D printing [16,30,31,32]. This technology allows for the fabrication of intricate structures with precise control over geometry, form and size, as well as mechanical and biological properties [33,34,36]. Various 3D printing techniques, such as high-precision SLA, DLP, FDM, and 2PP, have been employed to manufacture MNAs [40,44,53,107,108]. This section provides a distinctive evaluation of the advantages and limitations of fabrication techniques such as SLA, FDM, and 2PP for MNA production.

### 4.1. Stereolithography (SLA)

SLA is widely used for 3D printing of MNAs [30,33,37,38,39,40,41,42,43,44,45,46,47]. It is a 3D printing technology that fabricates parts in a layer-by-layer fashion using photochemical processes, by which UV light causes chemical monomers and oligomers to crosslink together to form polymers [109]. Solid objects of a curable material are printed in thin layers one on top of the other. A programmed movable spot beam of UV light shining on a surface or layer of UV-curable liquid is used to form a solid cross-section of the object at the surface of the liquid. The object is then moved, in a programmed manner, away from the liquid surface by the thickness of one layer, and the next cross-section is then formed and adhered to the immediately preceding layer defining the object. This process is continued until the entire object is formed [109]. Typical SLA processes exhibit superior resolution (10–150 μm) and surface roughness (0.38–0.61 μm) compared to some other techniques like material jetting, material extrusion, and powder bed fusion [37].

SLA printing has been used to create master molds for dissolvable PVP/polyvinyl alcohol (PVA) MNAs for ocular drug delivery [38]. It has been used to make coated MNAs out of a class I biocompatible resin, Dental SG, for intradermal insulin delivery [39] and to make hollow, dissolvable, and solid MNAs (made with materials like Dental SG; class IIa biocompatible resin Dental LT clear; and poly(propylene fumarate) (PPF) mixed with diethyl fumarate (DEF)) for transdermal drug delivery [9,40,42,43,110]. SLA MNAs have been fabricated using the commercial resin E-Shell 200 for transdermal electrochemical sensing [111] and as a base substrate for lab-on-a-microneedle devices made with VisiJet FTX Green, enabling the rapid detection of biomarkers in finger-prick blood samples [45]. It has been used to make hollow MNAs with the High Temp resin for blood-free detection of C-Reactive protein and Procalcitonin [46] in ISF. SLA-printed MNAs have been used to fabricate hollow MNAs for plant health monitoring using in situ electrochemical analysis by detecting biomarkers like H_2_O_2_, glucose, and pH [47]. Figure 6 shows two SLA 3D-printed MNAs [9,41].

### 4.2. Digital Light Processing (DLP)

DLP has been broadly used for 3D printing of MNAs [44,48,49,50,51,52]. DLP also uses photocurable resins to fabricate 3D parts, layer by layer, through spatially controlled solidification by using a projector light (either UV or white light) [112]. In addition, it is possible to tailor the final properties of the printed object by simply changing the photocurable resin formulations [113]. When light is projected onto the resin using DLP technology, instead of being restricted to a spot like in laser-assisted 3D printing, the entire layer is printed immediately as it uses a projector light. Hence, this technology allows fast printing [114,115]. The DLP printing technique has a relatively high resolution, which is usually at the micron scale [116]. The resolution of DLP 3D printing partially depends on the material chosen. For example, when the 3D constructs are printed using polyethylene glycol diacrylate (PEGDA MW = 700 da) solely, the XY resolution of the constructs can reach nearly 6 by 6 μm. Meanwhile, the resolution is about 17 μm with bioink containing 10% GelMA and 3 × 10^6^ cells/mL [117].

DLP-printed hollow MNAs made of PEGDA have been developed for transdermal drug delivery and multiplexed detection of biomarkers such as pH, glucose, and lactate in skin ISF [49]. Solid hydrogel MNAs manufactured with DLP have been developed for transdermal drug delivery [50]. Continuous glucose monitoring in ISF with sold MNAs made with biocompatible light-sensitive resins has also been successfully demonstrated by in vivo testing on mice [51]. They have also been used to develop drug delivery systems, utilizing hollow MNAs made with biocompatible class I resins (Dental SG), to the buccal tissue, as they increase the permeability of actives with molecular weights between 600 and 4000 Da [52].

Figure 7 shows a DLP 3D-printed MNA for detection and on-demand drug delivery applications on a wristband with its details [49].

### 4.3. Fused Deposition Modeling (FDM)

There are some issues with using FDM for 3D printing of MNAs, mainly due to its limited resolution. Nevertheless, it has been used in some studies to fabricate MNAs [53,54,55,56]. In this material extrusion process, a continuous filament of thermoplastic or composite material is used to construct 3D parts. The polymer filament is forced through a nozzle where the head is regulated by temperature, heated to the semi-liquid stage, and fed over the build plate or previously solidified substance. The product is built using a layer-by-layer technique at a constant speed and pressure [118]. Fused deposition modeling is a popular additive manufacturing technology because of its fast production, cost-efficiency, ease of access, broad material adaptation, and capability to produce complex components [118]. Acrylonitrile butadiene styrene (ABS), PLA, and polycarbonate (PC) are popularly used materials in FDM [118]. Post-processing is a vital process in FDM since the printed parts are not entirely ready for instant usage. After the printing process, the product is taken out from the bed platform, and the supporting structures are removed and undergo post-processing. This process is mainly used to improve the surface quality of the product [119]. In the context of 3D-printed MNAs, methods have been successfully developed that combine FDM with a post-fabrication etching step to yield ideally sized and shaped needles. These needles can insert, break off, and deliver small molecules into the skin without the need for a master template or mold [53].

As FDM-printed MNAs have been limited by their resolution and finish, they are less preferred for manufacturing MNAs. However, when combined with a post-fabrication chemical etching process, it has been shown that tip sizes as small as 1–55 μm can be achieved [53] (the typical resolution without post-fabrication steps is 50–200 μm [54]). FDM-printed biodegradable MNAs and BPMNAs made with PLA, with drugs absorbed into the polymer matrix (which can also be used as solid or coated MNAs), have been developed to load small molecules for transdermal drug delivery [53]. To facilitate high drug-loading capacity, BPMNAs with a drug reservoir made with PLA have been developed for transdermal drug delivery [55]. Coated PLA MNAs have also been developed for transdermal drug delivery [56].

### 4.4. Two-Photon Polymerization (2PP)

In recent years, 2PP has been developed to enable the manufacturing of elaborate structures in the micro- and nanoscale. In this specific technology, ultrashort laser pulses from a near-infrared femtosecond laser source are used to selectively polymerize photosensitive resins. The electronic excitation generated by the nearly simultaneous absorption of two photons is similar to that of a single photon with higher energy. This absorption provides a nonlinear energy distribution, centered at the laser’s focal point and with negligible absorption outside the immediate area of the laser’s focal volume. Upon absorption of this energy, photoinitiator molecules in the resin begin the polymerization process at locations known as “polymerization voxels”, where the energy surpasses a certain threshold. In comparison with other techniques, 2PP shows improved geometry control, as well as scalable resolution, reducing equipment, facilities, and maintenance costs commonly associated with etching and lithography-based methods. For this reason, researchers have exploited this technique to fabricate solid or hollow MN arrays using modified ceramics, inorganic-organic hybrid polymers, acrylate-based polymers, polyethylene glycol, and recently, water-soluble materials, with promising results [57].

A major advantage of 2PP is that it can achieve resolutions as low as 100 nm [58]. It has been used to mold dissolving and hydrogel-forming MNAs with an aqueous blend of PVP and PVA for controlled drug delivery of model drugs through skin models [57]. It has been used to develop hollow MNAs of organically modified ceramic hybrid materials (Ormocer^®^) for transdermal drug delivery [59], as well as to make hollow MNs with IP-S Photoresist for intracochlear diagnostics [60]. An interesting hybrid method of utilizing 2PP and electrochemical deposition was utilized to fabricate ultra-sharp gold-coated copper solid MNA for inner ear drug delivery [61]. It has been used to produce hollow MNAs with Eshell 300, a class 2a biocompatible material popular in the hearing aid industry, for transdermal sensing of electrolytes such as K+ ions [62]. Figure 8 shows some samples of 2PP 3D-printed MNAs. 

Table 1 provides an overview of the 3D printing techniques discussed for MNA fabrication, including SLA, DLP, FDM, and 2PP. It highlights the materials used for each method, the types of MNs produced, their advantages and disadvantages, and their applications.

## 5. Challenges and Limitations

This section examines the key challenges in the fabrication of 3D-printed MNAs, with a focus on material selection, mechanical strength, scalability, as well as regulatory and clinical considerations.

### 5.1. Material Selection and Optimization

Material selection is pivotal in the performance and safety of 3D-printed MNAs, as it directly influences both biocompatibility and mechanical properties. Achieving an optimal balance between these factors remains a significant challenge. Biocompatibility is essential, especially in medical applications, where materials must be non-toxic, non-immunogenic, and capable of integrating seamlessly with biological tissues. Common choices include biocompatible resins used in SLA and biodegradable polymers like PCL for drug delivery applications. However, these materials often face limitations in mechanical strength, which is critical for ensuring MNAs can reliably penetrate the skin without fracturing [120]. The trade-off between mechanical properties and printability poses another challenge. High-resolution techniques like SLA and 2PP require materials that can be precisely shaped while maintaining sufficient rigidity. Simultaneously, innovations in composite materials, combining polymers with nanoparticles or reinforcing fibers, are emerging to enhance both durability and functional performance. Furthermore, degradation and stability remain key considerations, particularly for dissolvable or biodegradable MNAs. Optimizing materials for predictable degradation profiles while preserving stability during storage is critical for effective drug delivery.

### 5.2. Mechanical Strength and Durability

The mechanical strength and durability of 3D-printed MNAs are critical to their effectiveness in applications ranging from transdermal drug delivery to biosensing. MNAs must withstand the forces involved in skin penetration while maintaining structural integrity. Mechanical failures, such as tip breakage or bending, can result in incomplete drug delivery or compromised sensor accuracy, posing significant risks in clinical settings [121]. Achieving the right balance between sharpness and strength is particularly challenging at the microscale. Techniques like SLA and 2PP enable high-resolution printing but often yield brittle structures due to the inherent properties of photopolymer resins. Researchers are addressing this limitation by developing composite materials and optimizing polymer formulations that offer improved toughness without sacrificing detail [82]. Durability is also crucial in applications where MNAs are expected to endure repeated insertions or prolonged exposure to biological environments. For instance, MNAs used for continuous monitoring or extended drug release must resist degradation while maintaining functionality. Advances in reinforcing MNAs with nanoparticles or integrating flexible polymers are promising strategies, but challenges remain in consistently achieving both strength and biocompatibility. As a result, the development of mechanically robust yet minimally invasive MNAs continues to be a focus of ongoing research and innovation.

### 5.3. Scalability and Manufacturing Cost

Despite the promising capabilities of 3D-printed MNAs, scaling their production for widespread clinical use remains a significant challenge. Fabrication methods like SLA and 2PP offer high resolution but are inherently limited by low throughput and high costs [122]. These techniques require specialized equipment and materials that drive up production expenses, making large-scale manufacturing difficult. For instance, 2PP can produce MNAs with nanoscale precision but is often prohibitively slow and expensive for commercial applications. Scalability also hinges on maintaining consistent quality as production volumes increase. Ensuring uniformity in MNA dimensions, tip sharpness, and material properties across batches is crucial for both safety and efficacy.

Also, scaling up the production of MNAs is challenging due to the absence of standardized manufacturing techniques and the complexities inherent in production. Many developers continue to use manual, lab-scale fabrication methods, which are inadequate for large-scale production. The diversity in MNA designs, formulations, and applications requires the creation of specialized equipment and innovative production processes. For instance, dissolvable MNAs need extended drying periods, making scaling even more difficult. Furthermore, the limited availability of contract manufacturing organizations (CMOs) capable of producing MNAs adds to the problem, as these companies are hesitant to invest in specialized machinery without proven demand, resulting in significant financial risks [123].

Additionally, the economic feasibility of 3D-printed MNAs is affected by material costs, which are often high for biocompatible resins and polymers tailored for MNA applications [124]. Cost-effective manufacturing solutions may involve hybrid approaches that combine 3D printing with other techniques or innovations that simplify post-processing steps. Overcoming these cost and scalability barriers is essential for transitioning MNAs from research prototypes to accessible medical devices in broader healthcare markets.

### 5.4. Regulatory and Clinical Considerations

Navigating regulatory concerns is equally challenging for the MNA industry, particularly due to the lack of established guidelines and precedents for this technology. Regulatory bodies have yet to clearly define the sterility requirements for MNAs, which fall between traditional transdermal patches and injectable technologies. The uncertainty surrounding these requirements poses significant risks for developers, who must choose between costly aseptic manufacturing or a potentially non-compliant low-bioburden process. Furthermore, quality control methods are still under development, with a need for technological advancements in non-destructive in-line quality control to ensure consistent product quality. The absence of standardized regulatory pathways hinders the industry’s ability to achieve widespread approval and commercial production of MNAs [121].

At present, the licensing process for MNA products is conducted on a per-application basis rather than for specific MNA systems (product-specific approval). Consequently, this fragmented approach to licensing leads to delays, hindering the commercialization of MNAs. To tackle this issue, a comprehensive regulatory framework for MNA-based licensing is necessary, encompassing aspects such as shape, formulation, sterilization, and packaging. By integrating current Good Manufacturing Practice (cGMP) standards and quality control measures, a licensing approach for MNAs based on quality by design should be implemented. This strategic shift aims to facilitate the commercialization of MNA products as pharmaceuticals [125].

Despite promising laboratory results, limited long-term clinical data are available for many 3D-printed MNAs, making it difficult to gain regulatory approval and build confidence among healthcare providers. Large-scale clinical trials are needed to assess both short-term and long-term outcomes, including biocompatibility, patient safety, and device durability. Additionally, the absence of standardized testing protocols for evaluating MNAs’ mechanical strength, drug delivery efficiency, and biodegradability further complicates the regulatory approval process. Addressing these challenges through collaboration between researchers, manufacturers, and regulatory bodies will be key to accelerating the development and market adoption of 3D-printed MNAs.

### 5.5. Skin Irritation

Another significant challenge in using MNAs is the potential for skin irritation. MNAs, while highly promising for applications such as targeted drug delivery, biosensing, and cosmetics, can induce irritation due to their mechanical disruption of the epidermis, activation of local inflammatory pathways, and variations in individual skin sensitivity. Irritation levels can be influenced by several factors, including the composition and design of the MNA, the materials used, and the nature of the active compounds delivered. Moreover, the lack of standardized methods for assessing and quantifying irritation presents a hurdle for consistent safety evaluations. To overcome these challenges, efforts must focus on optimizing MNA designs, employing advanced biocompatible materials, and refining delivery protocols to ensure minimal irritation without compromising efficacy. Incorporating personalized approaches tailored to individual skin types can further reduce irritation risks, enhance user satisfaction, and broaden the scope of MNA applications for long-term use [126].

### 5.6. Optimization of Microneedle Array Sizes

The optimization of MNA dimensions focuses on balancing geometry, needle height, tip radius, base diameter, and needle density to ensure effective skin penetration. Factors such as excessive needle length or insufficient spacing can hinder insertion efficiency or increase pain due to nerve contact. Additionally, the materials used in MNA fabrication must resist buckling or bending under insertion forces. Studies suggest that overly dense arrays can result in a “bed-of-nails” effect, reducing their effectiveness by distributing force inefficiently. Computational models and skin simulations play a critical role in optimizing these parameters to ensure uniform needle penetration without structural failure, a key requirement for clinical applications. Tailored designs are essential to enhance therapeutic outcomes while minimizing patient discomfort [120].

Research also demonstrates the significant role of MNA shape in determining skin penetration and transdermal drug administration efficacy. Polygonal-base MNAs with more vertices, such as star-shaped microneedles, exhibit superior penetration efficiency compared to circular designs. These star-shaped designs create deeper microchannels and require less force for insertion, improving drug release rates and quantities. For example, star-shaped MNAs have been shown to achieve up to 60% penetration depth and release drugs at rates 1.5–2 times higher than circular MNAs over 24 h. This shape-dependent performance underscores the potential of star-shaped MNAs in achieving efficient and controlled drug delivery for clinical applications [127].

The dimensions of MNAs also play a crucial role in their effectiveness for specific applications. Achieving optimal sizes is necessary to ensure efficient skin penetration while minimizing pain and irritation. Recent advancements in 3D printing technologies have facilitated precise control over MNA dimensions, enabling more consistent production and experimentation with new designs. However, challenges remain in achieving size uniformity during large-scale production. These challenges highlight the ongoing need for innovations in high-precision manufacturing techniques to ensure the scalability and reliability of MNAs for clinical and commercial use.

### 5.7. Drug Loading and Release

Effective drug loading and controlled release remain critical challenges in the development of MNAs. Achieving high drug-loading efficiency within the microneedle structures is essential, as these devices must maintain sufficient mechanical strength to penetrate the skin without breaking or bending. For hydrophobic and long-acting formulations, precise placement of the drug within the microneedle projections is necessary to minimize wastage and maximize delivery efficiency. Skin elasticity further complicates consistent insertion, potentially leading to incomplete drug deposition and reduced therapeutic efficacy. Controlled and sustained drug release relies heavily on the careful selection of materials. Biodegradable polymers such as PLA and PLGA are commonly used to facilitate gradual release through polymer degradation, ensuring prolonged therapeutic effects. Advanced manufacturing approaches, including embedding nanoparticles or utilizing layer-by-layer coating, are critical to achieving uniform drug distribution and controlled release kinetics. Optimizing microneedle geometry for drug capacity, mechanical strength, insertion efficiency, and biocompatibility remains crucial to advancing MNAs for broad clinical applications [128].

## 6. Conclusions and Future Perspectives

The field of 3D-printed MNAs is on the verge of transformative advancements, driven by innovations in material science, integration with complementary technologies, personalized medicine, and a focus on sustainability. These trends are poised to enhance both the performance and biocompatibility of MNAs, expanding their applications across medical and environmental domains.

One of the most significant areas of progress is in material science. Ongoing efforts are developing more sophisticated, biocompatible, and functional materials that improve key attributes of MNAs, such as mechanical strength, precise needle geometry, and optimized drug delivery. Multi-material printing, which combines materials with different properties in a single MNA, is emerging as a promising approach. This could lead to MNAs capable of performing multiple functions, such as controlled drug release and real-time biomarker monitoring, enhancing both durability and functionality.

The trend toward personalized medicine further shapes the future of 3D-printed MNAs. Advanced 3D printing techniques now enable the creation of customized MNAs tailored to individual patients’ needs. Personalization could involve adjusting the size, shape, and material composition of the arrays to suit a patient’s specific physiology and therapeutic requirements. These custom devices could improve treatment efficacy by maximizing drug absorption and reducing adverse reactions. The ability to rapidly prototype and produce these MNAs also paves the way for more accessible and cost-effective personalized therapies.

As the use of 3D-printed MNAs grows, sustainability considerations become increasingly important. Researchers are focusing on developing biodegradable and eco-friendly materials that reduce the environmental impact of medical waste. These materials not only contribute to sustainability but also enhance patient safety and comfort. Additionally, optimizing the 3D printing process to minimize material waste and energy consumption is critical for achieving more sustainable production methods. Future research may also explore the use of renewable resources in material synthesis, aligning MNA development with global sustainability goals.

While current material innovations have already significantly improved the safety and functionality of MNAs, future research will likely focus on developing novel materials with even greater multifunctional capabilities. For example, stimuli-responsive materials that react to environmental changes, such as pH or temperature, could enable on-demand drug release, further improving patient outcomes. The discovery and development of such materials could revolutionize the use of MNAs in medical treatments.

In conclusion, the future of 3D-printed MNAs is filled with exciting possibilities. By advancing material science, integrating emerging technologies, and adopting sustainable practices, researchers and developers can continue to push the boundaries of what is possible. The collaboration between material scientists, engineers, and healthcare professionals will be critical in transforming these advancements into reality, ultimately improving patient care and quality of life around the world.

## Figures and Tables

**Figure 1 micromachines-15-01433-f001:**
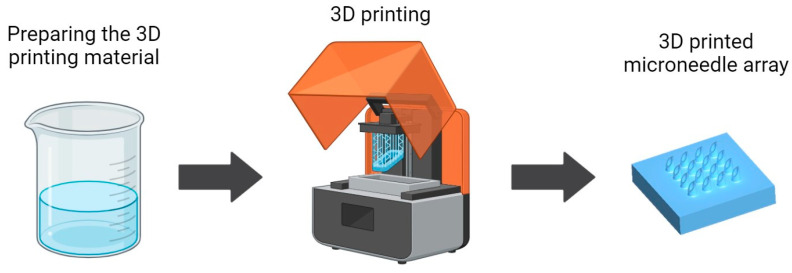
A schematic illustrating the 3D printing (resin-based) process for fabricating MNAs.

**Figure 2 micromachines-15-01433-f002:**
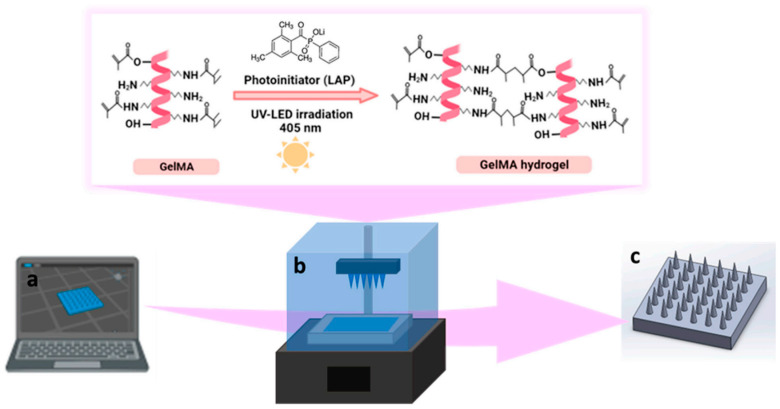
Schematic representation of the fabrication of MNAs using GelMA UV-curable resin; (**a**) designing of CAD model; (**b**) photocrosslinking of GelMA hydrogel during DLP printing (The top detail illustrates the photocrosslinking of GelMA, resulting in the formation of GelMA hydrogel); (**c**) 3D-printed GelMA MNA. The pink arrow depicts the flow of the process, starting with the CAD model design and concluding with the completed 3D-printed MNA. Reprinted under the terms of the Creative Commons Attribution License (CC BY) [67].

**Figure 3 micromachines-15-01433-f003:**
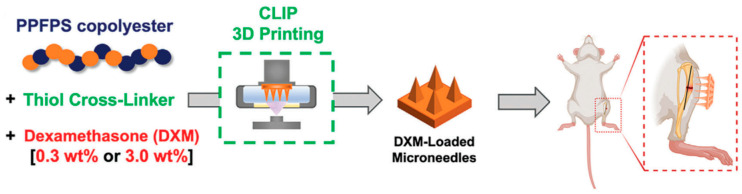
A schematic workflow for the fabrication of MNAs and its in vivo drug delivery evaluation. Reprinted with the permission of John Wiley and Sons [82].

**Figure 4 micromachines-15-01433-f004:**
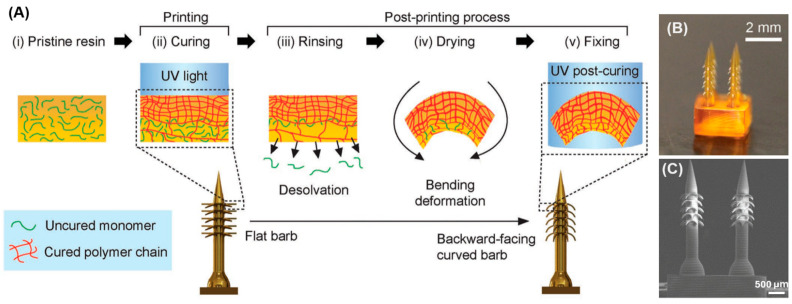
(**A**) Schematic illustration of 4D printing approach to program deformation of horizontally printed barbs into a backward-facing shape. Curing light initiates photopolymerization at the precursor solution’s surface (i), creating a gradient of cross-linking density as light intensity decreases through the solution (ii). During 3D printing, barbs remain horizontal, but rinsing in ethanol diffuses uncured monomers, leading to shrinkage and barb bending during drying (iii, iv). Postcuring with UV exposure fixes the curved barb shape (v); (**B**) Image of 4D-printed MNA with backward-facing barbs; (**C**) SEM image of 4D-printed MNA with backward-facing barbs. Reprinted with permission from Wiley [88].

**Figure 5 micromachines-15-01433-f005:**
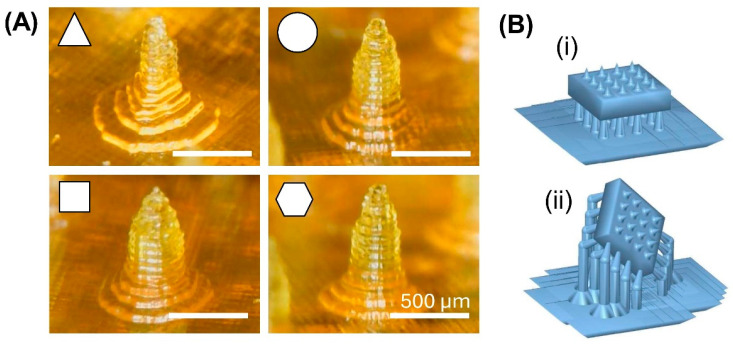
(**A**) Image of 3D-printed PEGDA-based MNAs with varying needle cross-sectional shapes, designed to evaluate the impact of needle shape on mechanical properties. Reprinted with permission from MDPI [12]. (**B**) Two designs for 3D-printed MNAs with different tilt angles: (i) tilt angle = 0° and (ii) tilt angle = 45° in two directions. Reprinted under the terms of the Creative Commons Attribution License (CC BY) [96].

**Figure 6 micromachines-15-01433-f006:**
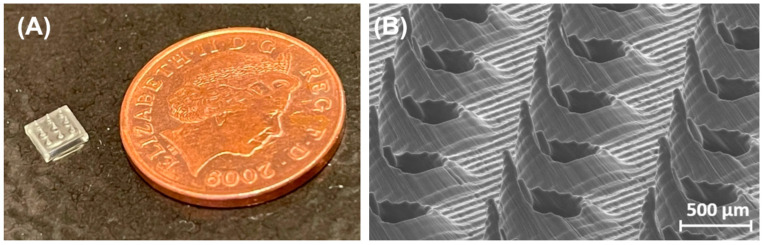
(**A**) SLA 3D-printed MNA compared to British one penny coin (diameter = 20.3 mm); Reprinted with permission from Elsevier [38]. (**B**) SEM image of the fine-tip microneedle array. Reprinted under the terms of the Creative Commons Attribution License (CC BY) [41].

**Figure 7 micromachines-15-01433-f007:**
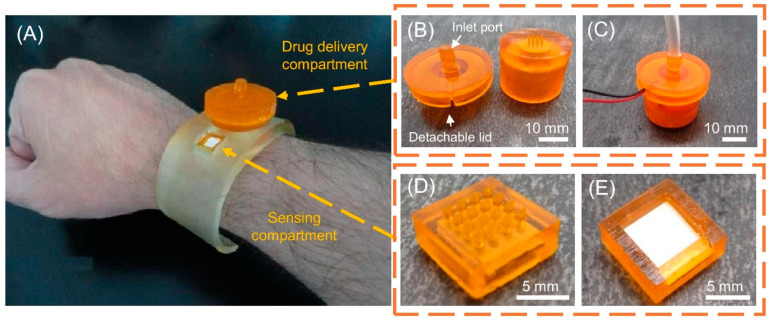
(**A**) DLP 3D-printed MNA for detection and on-demand drug delivery applications on a wristband; (**B**) disassembled drug delivery compartment; (**C**) assembled drug delivery compartment; (**D**) bottom view of the detection compartment; (**E**) top view of the detection compartment. Reprinted under the terms of the Creative Commons Attribution License (CC BY) [49].

**Figure 8 micromachines-15-01433-f008:**
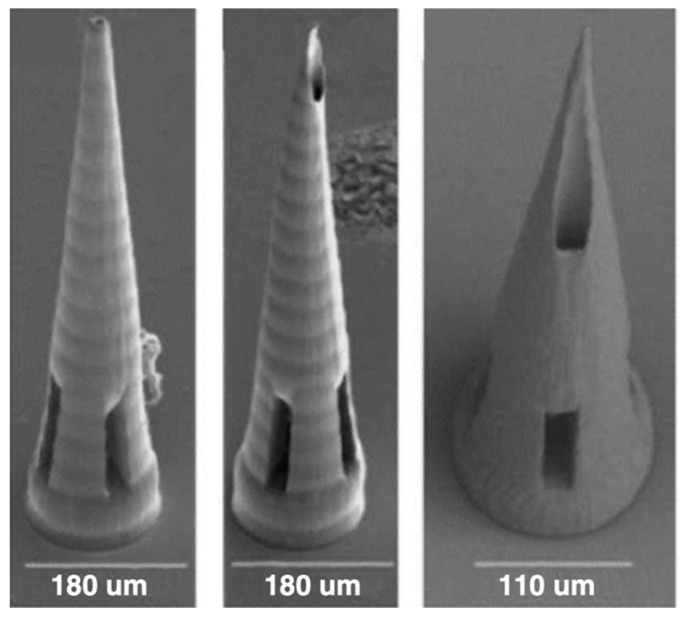
Samples of 2PP 3D-printed MNAs. Reprinted under a Creative Commons Attribution 4.0 International License [58].

**Table 1 micromachines-15-01433-t001:** Summary of 3D printing methods for MNA fabrication and their applications.

Fabrication Technique	Materials Used	MN Type	Advantages	Disadvantages	Applications of 3D-Printed MNAs	References
SLA	PVP/PVA. Dental SG. Dental LT clear. PPF mixed with DEF. e-Shell 200. VisiJet FTX Green. High Temp resin.	Hydrogel, Coated, Hollow, Dissolvable, Solid	High resolution	Relatively expensive. Slow process. Limited materials	Transdermal drug delivery. Ocular drug delivery. Interdermal drug delivery. Intercochlear diagnostics. Transdermal electrochemical sensing. Biomarker detection in blood. Blood free detection of proteins in ISF. Plant health monitoring.	[30,33,37,38,39,40,41,42,43,44,45,46,47]
DLP	PEGDA. Biocompatible light sensitive resins. Dental SG.	Hollow, Hydrogel, Solid, Coated.	Fairly high resolution	Moderate to expensive. Lower resolution and surface finish compared to SLA	Transdermal drug delivery. Drug delivery through buccal tissue. Multiplexed biomarker detection in ISF. Continuous glucose monitoring.	[44,48,49,50,51,52]
FDM	PLA	Solid, Coated, Dissolvable.	Affordable. High throughput.	Low resolution. Often requires post-fabrication processes	Transdermal drug delivery	[53,54,55,56]
2PP	PVP/PVA. Ormocer. IP-S Photoresist coated with gold. Eshell 300	Hollow, Hydrogel, Dissolvable, Solid.	Very high resolution	Expensive. Slow process	Transdermal drug delivery. Intracochlear diagnostics. Inner ear drug delivery. Transdermal sensing of electrolytes.	[57,58,59,60,61,62]

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
