# Peer review of "Advancements in Materials for 3D-Printed Microneedle Arrays: Enhancing Performance and Biocompatibility"

_micromachines, 2024, doi:10.3390/mi15121433_

Round 1
Reviewer 1 Report
Comments and Suggestions for Authors
The authors have compiled a good overview of the topic. However, here are my few concerns:
1. The introduction can be made more comprehensive by citing results from recent studies.
2. Several reviews are available, which have explained the topic similarly. The authors are suggested to highlight the review's uniqueness and major novel points in every section.
3. The details under different subsections of section 2—types of materials—are not explained in a similar way. Some are detailed, while others are abruptly ended. Please take care of this.
4. Sections 3.2 and 3.3. on biocompatibility, degradation, and stability need to be elaborated.
Author Response
- The introduction can be made more comprehensive by citing results from recent studies.
Response: Thank you for your valuable feedback. We have incorporated the results of some recent work into the introduction section.
[Page 1]
For instance, dissolving MNAs have been demonstrated to deliver monoclonal antibodies more effectively than subcutaneous injections, showing enhanced therapeutic effects in osteoporosis models [5].
[Page 1]
Notably, customizable photothermal MNAs produced using low-cost 3D printers have been shown to effectively treat bacterial infections via intradermal photothermal therapy [15].
[Page 2]
For example, high-resolution 3D printing has enabled the production of latticed MNAs with enhanced cargo loading and tunable release kinetics for therapeutic applications [29].
[Page 2]
Recent work highlights that the use of biocompatible photocurable resins in MNAs not only provides mechanical durability but also reduces environmental impact, aligning with sustainability goals [5].
[Page 2]
For instance, gelatin methacryloyl (GelMA)-based MNAs produced using vat photopolymerization demonstrated high-resolution geometries and effective transdermal drug delivery with minimal mechanical failure [67].
- Several reviews are available, which have explained the topic similarly. The authors are suggested to highlight the review's uniqueness and major novel points in every section.
Response: Thank you for your insightful feedback. We have added sentences across various sections to emphasize the uniqueness of the review paper and its key novel points.
[Page 1- Abstract]
Unlike previous reviews, this paper emphasizes the integration of innovative materials with advanced 3D printing techniques to enhance both the performance and sustainability of MNAs.
[Page 2- Introduction]
While earlier reviews have explored the general advancements in 3D-printed MNAs, this review uniquely focuses on the intersection of material science and emerging trends like personalized medicine and stimuli-responsive materials.
[Page 3]
This review examines a variety of materials, including photopolymer resins, biodegradable polymers, hydrogels, and emerging materials, used in the fabrication of 3D-printed MNAs, with a focus on addressing underexplored aspects highlighted in previous reviews.
[Page 10]
This section provides a distinctive evaluation of the advantages and limitations of fabrication techniques such as SLA, FDM, and 2PP for MNA production.
[Page 14]
This section examines the key challenges in the fabrication of 3D-printed MNAs, with a focus on material selection, mechanical strength, scalability, as well as regulatory and clinical considerations.
- The details under different subsections of section 2—types of materials—are not explained in a similar way. Some are detailed, while others are abruptly ended. Please take care of this.
Response: Thank you for your valuable comment. We have added some additional details to address the issue and ensure consistency in the explanations.
[Pages 5]
As mentioned, the structure of hydrogels can absorb and retain water, enabling efficient encapsulation and controlled release of drugs. This makes them ideal for transdermal drug delivery, ensuring minimal discomfort and sustained therapeutic effects. Furthermore, their biocompatibility reduces adverse reactions, enhancing patient safety [83]. In biosensing, hydrogels’ ability to integrate with biomolecules and respond to environmental stimuli, such as pH or glucose levels, allows precise, real-time monitoring, making them indispensable in advanced biomedical applications [12].
Research demonstrated that 3D-printed gelatin GelMA hydrogel MNAs possess customizable heights and sharpness, achieving exceptional mechanical performance without breakage under a displacement of up to 0.3 mm. These findings highlight GelMA MNAs as mechanically robust and highly suitable for transdermal drug delivery, providing precise control over geometry and mechanical properties with high-resolution fabrication [67].
[Pages 5, 6]
A recent study [92] presented an innovative 3D-printed MNA utilizing a compo-site hydrogel ink made from four monomers: 2-(dimethylamino)ethyl methacrylate, N-isopropylacrylamide, acrylic acid, and acrylamide. These monomers were cross-linked using aluminum hydroxide nanoparticles, yielding a material with outstanding mechanical strength suitable for precise 3D printing. The printed MNAs showcased customizable geometries and triple responsiveness to pH, temperature, and glucose. Moreover, their effectiveness for cytotoxicity-free transdermal drug delivery was confirmed using bovine serum albumin as a model drug. This innovative approach highlights the potential of advanced composite formulations for developing multifunction-al MNAs in biomedical applications.
- Sections 3.2 and 3.3. on biocompatibility, degradation, and stability need to be elaborated.
Response: Thank you for your valuable feedback. We have added further discussion to Sections 3.2 and 3.3 to elaborate on biocompatibility, degradation, and stability.
[Pages 8, 9]
One of the significant challenges in achieving biocompatibility of 3D-printed MNAs lies in the presence of residual monomers and photoinitiators after the 3D printing process. These unreacted components can potentially cause cytotoxic effects, posing risks to the safety and efficacy of the MNAs in biomedical applications [76]. To overcome this issue, thorough post-processing techniques are essential. These methods aim to remove any residual monomers and photoinitiators, thereby minimizing cytotoxicity and enhancing the biocompatibility of the final product. Strategies such as solvent washing, UV curing, and thermal treatments have been explored to effectively mitigate this challenge.
[Pages 8, 9]
Tailored degradation rates are essential for MNA materials to align with application needs. Rapid degradation is ideal for short-term drug release, ensuring timely therapeutic effects, while slow, controlled degradation supports sustained delivery over extended periods. Customizing degradation profiles enhances efficacy, minimizes side effects, and optimizes treatment outcomes for diverse medical applications. The use of hydrogels and biodegradable polymers allows precise control over degradation, aligning material properties with specific medical needs for diverse treatment scenarios [83].

Reviewer 2 Report
Comments and Suggestions for Authors
Comments: micromachines-3292614
The article looks interesting and informative to me. The authors addressed major concerns of the microneedles array in their manuscript. However, I have a few comments to improve their manuscript as follows.
1. In the abstract, it is not clear what new concepts they brought into their manuscript. Please address this issue.
2. In the “Challenges and Limitations” section, it would be nice to see to see how the deal with the skin irritations by the advancement of the microneedles.
3. The authors discussed about the optimization of the materials in the challenges section. It would be nice to see a detailed discussion on optimization of the sizes.
4. There are still issues with the drug loading and release. It is suggested to discuss about them in the challenges section.
Comments on the Quality of English LanguageIt's good, but still scope to improve the quality.
Author Response
- In the abstract, it is not clear what new concepts they brought into their manuscript. Please address this issue.
Response: Thank you for your insightful feedback. We have revised the abstract to clarify the new concepts introduced in the paper.
Abstract: The rapid advancement of 3D printing technology has revolutionized the fabrication of microneedle arrays (MNAs), which hold great promise in biomedical applications such as drug delivery, diagnostics, and therapeutic interventions. This review uniquely explores advanced materials used in the production of 3D-printed MNAs, including photopolymer resins, biocompatible materials, and composite resins, designed to improve mechanical properties, biocompatibility, and functional performance. Additionally, it introduces emerging trends such as 4D printing for programmable MNAs. By analyzing recent innovations, this review identifies critical challenges and proposes future directions to advance the field of 3D-printed MNAs. Unlike previous reviews, this paper emphasizes the integration of innovative materials with advanced 3D printing techniques to enhance both the performance and sustainability of MNAs.
- In the “Challenges and Limitations” section, it would be nice to see to see how the deal with the skin irritations by the advancement of the microneedles.
Response: Thank you for your valuable comment. We have incorporated a subsection in the "Challenges and Limitations" section discussing the issue of skin irritation associated with the use of microneedle arrays.
[Pages 15, 16]
5.5. Skin irritation
Another significant challenge in using MNAs is the potential for skin irritation. MNAs, while highly promising for applications such as targeted drug delivery, bio-sensing, and cosmetics, can induce irritation due to their mechanical disruption of the epidermis, activation of local inflammatory pathways, and variations in individual skin sensitivity. Irritation levels can be influenced by several factors, including the composition and design of the MNA, the materials used, and the nature of the active compounds delivered. Moreover, the lack of standardized methods for assessing and quantifying irritation presents a hurdle for consistent safety evaluations. To overcome these challenges, efforts must focus on optimizing MNA designs, employing advanced biocompatible materials, and refining delivery protocols to ensure minimal irritation without compromising efficacy. Incorporating personalized approaches tailored to individual skin types can further reduce irritation risks, enhance user satisfaction, and broaden the scope of MNA applications for long-term use [127].
- The authors discussed about the optimization of the materials in the challenges section. It would be nice to see a detailed discussion on optimization of the sizes.
Response: Thank you for your valuable comment. We have added a subsection to the "Challenges and Limitations" section to discuss the optimization of microneedle array sizes.
[Page 16]
5.6. Optimization of microneedle array sizes
The optimization of MNA dimensions focuses on balancing geometry, needle height, tip radius, base diameter, and needle density to ensure effective skin penetration. Factors such as excessive needle length or insufficient spacing can hinder insertion efficiency or increase pain due to nerve contact. Additionally, the materials used in MNA fabrication must resist buckling or bending under insertion forces. Studies suggest that overly dense arrays can result in a "bed-of-nails" effect, reducing their effectiveness by distributing force inefficiently. Computational models and skin simulations play a critical role in optimizing these parameters to ensure uniform needle penetration without structural failure, a key requirement for clinical applications. Tailored designs are essential to enhance therapeutic outcomes while minimizing patient discomfort [121].
Research also demonstrates the significant role of MNA shape in determining skin penetration and transdermal drug administration efficacy. Polygonal-base MNAs with more vertices, such as star-shaped microneedles, exhibit superior penetration efficiency compared to circular designs. These star-shaped designs create deeper microchannels and require less force for insertion, improving drug release rates and quantities. For example, star-shaped MNAs have been shown to achieve up to 60% penetration depth and release drugs at rates 1.5–2 times higher than circular MNAs over 24 hours. This shape-dependent performance underscores the potential of star-shaped MNAs in achieving efficient and controlled drug delivery for clinical applications [128].
The dimensions of MNAs also play a crucial role in their effectiveness for specific applications. Achieving optimal sizes is necessary to ensure efficient skin penetration while minimizing pain and irritation. Recent advancements in 3D printing technologies have facilitated precise control over MNA dimensions, enabling more consistent production and experimentation with new designs. However, challenges remain in achieving size uniformity during large-scale production. These challenges highlight the ongoing need for innovations in high-precision manufacturing techniques to ensure the scalability and reliability of MNAs for clinical and commercial use.
- There are still issues with the drug loading and release. It is suggested to discuss about them in the challenges section.
Response: Thank you for your insightful comment. We have included a new subsection in the "Challenges and Limitations" section to address the issues related to drug loading and release in MNAs.
[Pages 16, 7]
5.7. Drug loading and release
Effective drug loading and controlled release remain critical challenges in the development of MNAs. Achieving high drug-loading efficiency within the microneedle structures is essential, as these devices must maintain sufficient mechanical strength to penetrate the skin without breaking or bending. For hydrophobic and long-acting formulations, precise placement of the drug within the microneedle projections is necessary to minimize wastage and maximize delivery efficiency. Skin elasticity further complicates consistent insertion, potentially leading to incomplete drug deposition and reduced therapeutic efficacy. Controlled and sustained drug release relies heavily on the careful selection of materials. Biodegradable polymers such as PLA and PLGA are commonly used to facilitate gradual release through polymer degradation, ensuring prolonged therapeutic effects. Advanced manufacturing approaches, including embedding nanoparticles or utilizing layer-by-layer coating, are critical to achieving uniform drug distribution and controlled release kinetics. Optimizing microneedle geometry for drug capacity, mechanical strength, insertion efficiency, and biocompatibility remains crucial to advancing MNAs for broad clinical applications [129].
